# Evaluation of Linear Programming and Optimal Contribution Selection Approaches for Long-Term Selection on Beef Cattle Breeding

**DOI:** 10.3390/biology12091157

**Published:** 2023-08-23

**Authors:** Xu Zheng, Tianzhen Wang, Qunhao Niu, Jiayuan Wu, Zhida Zhao, Huijiang Gao, Junya Li, Lingyang Xu

**Affiliations:** State Key Laboratory of Animal Biotech Breeding, Institute of Animal Science, Chinese Academy of Agricultural Sciences, Beijing 100193, China; 82101202357@caas.cn (X.Z.); 82101201172@caas.cn (T.W.); 82101211643@caas.cn (Q.N.); 82101215391@caas.cn (J.W.); 82101225461@caas.cn (Z.Z.); gaohuijiang@caas.cn (H.G.)

**Keywords:** simulation, linear programming, optimal contribution selection, genetic gain, average kinship coefficient, cattle breeding

## Abstract

**Simple Summary:**

The effect of optimized mating methods for long-term selection has not been studied in cattle breeding. In this study, the linear programming and optimal contribution selection methods on the genetic gain and inbreeding level of beef cattle were explored and evaluated using a simulation strategy. Our results showed that the linear programming method can effectively improve the genetic gain in the population during long-term selection in the breeding process, and the optimal contribution selection method can maintain a balance between improving genetic gain and controlling inbreeding level. Our findings can provide theoretical guidance for the long-term and sustainable genetic gain in breeding populations in farm animals.

**Abstract:**

The optimized selection method can maximize the genetic gain in offspring under the premise of controlling the inbreeding level of the population. At present, genetic gain has been largely improved by using genomic selection in multiple farm animals. However, the design of the optimal selection method and assessment of its effects during long-term selection in beef cattle breeding are yet to be fully explored. In this study, a simulated beef cattle population was constructed, and 15 generations of simulated breeding were carried out using the linear programming breeding strategy (LP) and optimal contribution selection strategy (OCS), respectively. The truncation selection strategy (TS−I and TS−II) was used as the control. During the breeding process, genetic parameters including genetic gain, average kinship coefficient, QTL effect variance, and average observed heterozygosity were calculated and compared across generations. Our results showed that the LP method can significantly improve the genetic gain in the population, especially the genetic performance of the traits with high heritability and the traits with high weight in the breeding process, but the inbreeding level of the population is higher under LP strategy. Although the genetic gain in the population under the OCS strategy is lower than the TS−II strategy, this method can effectively control the inbreeding level of the population. Our findings also suggest that the LP and OCS method can be used as an effective means to improve genetic gain, while the OCS method is a more ideal method to obtain sustainable genetic gain during long-term selection.

## 1. Introduction

Genomic selection (GS) has been widely applied in livestock breeding in the past few decades [1,2,3,4,5]. GS has merits including increasing prediction accuracy, promoting genetic gain, and shortening generation interval [6,7,8].

Many studies have pointed out that GS can lead to the increase in inbreeding level of livestock population during long-term selection, which will reduce the genetic diversity of the population, slow down the growth in genetic gain, and even cause inbreeding depression [9,10,11,12]. Therefore, to achieve long-term and sustainable genetic gain, it is necessary to improve genetic gain while controlling the inbreeding level of the population by optimizing the mating during the breeding process [13,14,15].

Many optimized methods including the Linear Programming method (LP) and the Optimal Contribution Selection method (OCS) have been proposed in recent decades. The LP method is a branch of operational research [16]. The main steps for the application of LP in breeding include the following instructions [17]. Firstly, construct a selection index for each possible mating combination, then construct an appropriate objective function based on the selection index, and finally solve the mating combination that meets the requirements. 

The OCS method was proposed by Meuwissen [18] based on the theory of genetic contribution [19]. This method maximizes the weighted genetic value of parents, while also limiting the genetic relationships between parents and selecting the most suitable mating combination from them [20]. Previous studies have compared the application of GS and OCS breeding strategies using pig data, and their findings showed that the inbreeding level of the population under the OCS strategy was below 5%, while the inbreeding level under GS strategy was as high as 10~15% [21]. Pryce et al. [22] demonstrated that using inbreeding coefficients calculated based on genomic information in the OCS method can reduce the impact of inbreeding on genetic gain. Clark et al. [23] also revealed that incorporating genomic information can achieve higher genetic gain while controlling the same inbreeding level. Hjorto et al. used the OCS method to reduce the frequency of lethal recessive alleles within the population, and estimated true genetic gain realized at a 0.01 rate of true inbreeding. Their results showed that the true genetic gain increased 12% using the OCS strategy compared to truncation selection [24]. Kohl et al. [25] confirmed the feasibility of the OCS method using simulation technology in Vorderwald cattle breeding, they found that the advanced OCS facilitated the management of the migrant contributions and the rate of inbreeding at native alleles.

To achieve sustainable genetic gain under long-term selection, it is also necessary to consider the loss of rare favorable alleles. One recent study proposed a weighted genomic selection method by weighting the effects of rare favorable alleles, and achieved the goal of maintaining genetic diversity [26].

Most studies on OCS methods are aimed at maximizing genetic gain by controlling the inbreeding coefficient. Few studies exist on the OCS method that minimize the population inbreeding coefficient as a breeding objective. Also, the LP methods or the OCS method have not been fully studied as a breeding program in cattle. Therefore, the purpose of this study is to: (1) explore the application effect of the LP method in controlling inbreeding level and improving genetic gain in cattle breeding; (2) evaluate the effects of OCS strategies with different breeding objectives during long-term selection; (3) verify the feasibility of OCS strategy in achieving sustainable genetic gain during long-term selection process. Our study may provide theoretical guidance for sustainable genetic gain achieved during long-term selection process in farm animal breeding.

## 2. Materials and Methods

### 2.1. Generation of Simulated Population

We generated a simulated historical population that consisted of 1000 individuals (500 males and 500 females). The genome of each individual in the population contains 29 chromosomes with a length of 2.33 M. The number of SNPs per individual is 100 K, evenly arranged on chromosomes. A total of 750 QTLs were randomly arranged on chromosomes, including 125 dominant QTLs. Each individual has 5 simulated phenotypic traits with heritability of 0.44, 0.48, 0.45, 0.3, and 0.22 [27], and these traits were indicated as Trait 1, Trait 2, Trait 3, Trait 4, and Trait 5.

All males and females in the historical population randomly mated to produce a total of 1000 offspring (500 males and 500 females) as the 2nd generation population. Then, all the males and females also mated randomly to generate 1000 offspring. Using the same breeding strategy, a total of 1000 generations were generated.

The population from the 1st generation to the 1001st generation were generated by random selection to generate initial linkage disequilibrium (LD) and mutation–drift equilibrium, while the population size was 1000 individuals and the sex ratio was 1:1. From the 1001st generation to 1006th generation, the random selection strategy was used, and the population size was gradually expanded in the breeding process. The population size was 1400 individuals in 1006th generation. During this process, the proportion of females gradually increases, and the proportion of males gradually decreases. Finally, 200 male individuals and 1200 female individuals remained in the 1006th generation population. The 1006th generation population were divided evenly into two populations as the Founder population (FP1 and FP2). In the above breeding process, the mutation rate of alleles was 10^−5^.

### 2.2. Simulation of Different Breeding Strategies

We designed two types of optimized breeding strategies: (1) linear programming strategy (LP), (2) optimal contribution selection strategy (OCS). Two different truncation selection strategies (TS−I and TS−II) were designed. The TS−I is the control strategy for LP, focusing on the genetic gain in Trait 1 to Trait 4, while TS−II is the control strategy for OCS, focusing on the genetic gain in Trait 5.

#### 2.2.1. LP Strategy

The main breeding goal of the LP strategy is to improve the genetic gain in Trait 1 to Trait 4. First, we calculated the weighted sum of GEBV (WSG) of four traits (from Trait 1 to Trait 4) per individual in FP1:$${WSG}_{i}=0.35\times {GEBV}_{Trait1}+0.25\times {GEBV}_{Trait2}+0.25\times {GEBV}_{Trait3}+0.15\times {GEBV}_{Trait4}$$
where the ${WSG}_{i}$ is the WSG of individual i, the ${GEBV}_{Traitk}$ (*k* = 1, 2, 3, 4) are GEBVs of four traits of individual i which were calculated by the GBLUP method. We selected the top 10% male individuals and the top 50% female individuals with the highest WSG in the population as candidates. The set of weighting coefficients was according to the Genomic China Beef Index.

Then, we constructed a selection index for potential offspring of each mating, and the calculation formula is [28]:$$I_{ij}=\frac{{WSG}_{ij}-\gamma f_{ij}}{var\left(WSG\right)}$$
where the $I_{ij}$ is the selection index of the offspring of individual i and individual j, ${WSG}_{ij}$ is the expected WSG of the offspring of individual i and individual j, γ is the effect of inbreeding on WSG [29], $f_{ij}$ is the expected progeny inbreeding from that mating, $var\left(WSG\right)$ is the standard deviation of all candidates.

The aim of LP method is maximizing the sum of the selection index for the whole potential offspring:$$f_{\left(max\right)}\left(I\right)=\sum_{i=1}^{a}\sum_{j=1}^{b}I_{ij}x_{ij}$$
where the $I_{ij}$ is the selection index of the offspring of individual i and individual j, a is the number of male candidates, b is the number of female candidates, $x_{ij}$ is equal to 1 or 0 according to the following restrictions:$$\sum_{i=1}^{a}x_{ij}\leq 10;\sum_{j=1}^{b}x_{ij}\leq 1$$

These two restrictions mean that each male candidate can be mated to 10 females, while each female candidate can be mated only once at most.

Then, the next generation population was generated based on the calculated specific mating combinations. The same selection strategy was used to simulate the subsequent 14 generations. Population size and gender ratio are consistent with FP1 during breeding process.

#### 2.2.2. TS−I Strategy

We calculated the GEBVs for Trait 1 to Trait 4 of all individuals in FP1 using GBLUP method. Next, we calculated the WSG of all individuals. Then, the top 10% male individuals and top 50% female individuals with highest WSG were selected as candidates. All candidate individuals were randomly mated to produce 700 offspring as the next generation. 

The following 14 generations were simulated using the same selection strategy. The population size and gender ratio remained unchanged during the breeding period.

#### 2.2.3. OCS Strategies

Under this breeding strategy, Trait 5 of individuals within the population is considered as the target trait. We first selected the top 10% male individuals in FP2 that had the highest genetic contributions to the offspring population as candidates, and then selected different female candidates according to two different breeding targets: (1) set an upper bound for the increase in the kinship of the offspring population and maximize the BVE of the offspring population under this upper bound. This breeding strategy is recorded as OCS_maxBVE; (2) set a certain lower bound for the BVE increase in the offspring population and minimize the kinship of the offspring population on the basis of meeting the lower bound. The lower bounds of BVE increase were set to four levels: 0.05, 0.07, 0.1, and 0.2, and the corresponding breeding strategies are recorded as OCS_minKin I, OCS_minKin II, OCS_minKin III and, OCS_minKin IV, respectively.

Similarly, the mating of candidate individuals produces 700 offspring to maintain the population size, with a gender ratio of 1:6. Subsequently, the same breeding strategy was used to simulate the generation of the subsequent 14 generations.

#### 2.2.4. TS−II Strategy

We calculated the GEBV of Trait 5 for all individuals in FP2 using GBLUP method. Then, we selected the individuals with the highest GEBV as a candidate with the same selection intensity as LP and OCS strategies. All candidate individuals are randomly mated to produce 700 offspring.

The next 14 generations were simulated using the same selection strategy, and the population size and gender ratio of each generation remained unchanged.

The generation of the founder populations and the simulated breeding process under various breeding strategies were shown in Figure 1.

### 2.3. Calculation of Genetic Parameters

We calculated the following genetic parameters to compare different selection strategies in breeding simulation.

#### 2.3.1. Genetic Gain

The difference between the average GEBV of all individuals in each generation and the average GEBV of all individuals in the FP is the genetic gain in the corresponding traits in the current generation. The calculation formula is as follows [30]:$${\mathrm{Genetic}\;\mathrm{gain}}_{ij}=\frac{1}{M}\sum_{k=1}^{M}\left(\frac{\sum_{l=1}^{N}{GEBV}_{ijkl}}{N}-\frac{\sum_{l=1}^{N}{GEBV}_{FPjkl}}{N}\right)$$
where the ${\mathrm{Genetic}\;\mathrm{gain}}_{ij}$ is genetic gain in the jth trait in the ith generation, k represents the repetitions, l represents the number of individuals of the population, M represents the repetition times, and N represents the population size.

#### 2.3.2. Average Kinship Coefficient

The kinship coefficient between individual *i* and individual *j* is calculated as follows:$${\mathrm{Kinship}}_{ij}=\frac{1}{M}\sum_{k=1}^{M}\left[\frac{f_{IIS_{ijk}}-p_{k}^{2}-q_{k}^{2}}{2p_{k}q_{k}}\right]$$
where *M* is the number of SNPs on the simulated genomes, *p_k_* and *q_k_* are the frequencies of the two alleles at locus *k* for the current generation, and $f_{IIS_{ijk}}$ is the probability that the two alleles are identical in state (IIS) [21]. Then, we calculated the average of the kinship coefficients of all individuals within each generation as the average kinship coefficient of that generation to represent the inbreeding level.

#### 2.3.3. QTL Effect Variance and Average Observed Heterozygosity

The QTL effect variances and averages observed heterozygosity of the FP and the 1021st generation populations were calculated to evaluate the degree of change in genetic diversity before and after the breeding process [31]. 

The observed heterozygosity quantifies the amount of genetic variation caused by polymorphic loci [32]. We used software PLINK v1.90 to calculate the average observed heterozygosity for each breeding scenario.

The QTL effect variances were calculated using the following formula:$$\sigma^{2}=\frac{1}{n}\sum_{i=1}^{n}2p_{i}\left(1-p_{i}\right)a_{i}^{2}$$
where the n is the number of the QTLs, $p_{i}$ is the allele frequency at locus i, $a_{i}$ is the effect of the QTL i.

### 2.4. Software

The generation of the FP, simulation of phenotype and genotype, and creation of new generations were completed through R package MoBPS (https://cran.r-project.org/package=MoBPS (accessed on 1 July 2023)). The GEBV of individual was calculated using R package emmreml (https://cran.r-project.org/package=EMMREML (accessed on 1 July 2023)). The solution of linear programming problem used R package lpSolve (https://cran.r-project.org/package=lpSolve (accessed on 1 July 2023)). The implementation of the OCS method called the R package optiSel (https://cran.r-project.org/package=optiSel (accessed on 1 July 2023)).

## 3. Results

In this study, we utilized the simulation method to construct breeding populations in cattle and conducted a breeding program for 15 generations under three breeding strategies including LP, OCS, and TS. Four genetic parameters including genetic gain, average kinship coefficient, QTL effect variance, and average observed heterozygosity were estimated during the process to evaluate the role of LP and OCS methods in improving genetic gain and controlling population inbreeding levels.

### 3.1. Genetic Gain

The genetic gains of four traits (from Trait 1 to Trait 4) in populations under LP and TS−I breeding strategies increase with generations, as shown in Figure 2. The genetic gain in Trait 1 was the highest and that of Trait 4 was the lowest under the two breeding strategies. Under the TS−I breeding strategy, the genetic gain in Trait 2 in the population was higher than that of Trait 3 before the 3rd generation; however, the genetic gain in Trait 3 gradually exceeded that of Trait 2 after the 3rd generation. In contrast, the genetic gain in Trait 2 was constantly higher than that of Trait 3 under the LP strategy.

As for the WSG, this value increases with generations under LP and TS−I strategies. Under the LP strategy, the average WSG of the 1021st generation population is 3.411 higher than the FP1. Under the TS−I strategy, the average WSG of the 1021st generation population is 3.362 higher than the FP1.

In this study, we found the genetic gain in the same trait varies under different breeding strategies (Figure 3). LP strategy has a higher degree of improvement in genetic gain in Trait 1 and Trait 2 than the TS−I strategy. The genetic gains of Trait 1 and Trait 2 of the last generation were 4.36 and 3.78 under the LP strategy, and 4.07 and 3.21 under the TS−I strategy, respectively. The genetic gain in the first three generations of Trait 3 under the LP strategy is slightly higher than that of the TS−I strategy. After the 3rd generation, the genetic gain in this trait under the LP strategy is lower, and ultimately is 3.50 and 3.79, respectively. However, the final genetic gain in Trait 4 under the LP strategy is lower than that under the TS−I strategy. The genetic gain in this trait under the LP strategy is ultimately 2.35, and under the TS−I strategy is 2.50.

Similarly, the TS−II strategy and the five OCS breeding strategies can significantly improve the genetic progress of the population (Figure 4A). The TS−II strategy has the highest improvement in genetic gain, ultimately reaching 6.51. The genetic gains of the population at the last generation under all OCS strategies are lower than that of the TS−II strategy, but the gap between them is not significant. The genetic gain is 5.19 under OCS_maxBVE strategy, 5.09 under OCS_minKin I strategy, 5.00 under OCS_minKin II strategy, 5.36 under OCS_minKin III strategy, and 5.35 under OCS_minKin IV strategy.

### 3.2. Average Kinship Coefficient

Due to the existence of selection, the average kinship coefficient of the population gradually increases with generations. The average kinship coefficient of the 1021st generation under LP strategy is higher than that under TS−I strategy, which is 0.31 and 0.28, respectively (Figure 5).

All OCS strategies can effectively control the inbreeding than TS−II strategies based on the estimation of average kinship coefficient across generations (Figure 4B). The average kinship coefficient of the population under the TS−II strategy is ultimately 0.36. The control effect of OCS_maxBVE strategy on population average kinship coefficient is the best among all OCS strategies, and the final result is 0.19. The population average kinship coefficient under the four OCS strategies that minimize kinship is slightly higher than the OCS_maxBVE strategy, and the higher the BV increase lower bound for the OCS_minKin strategy, the higher the final population average kinship coefficient. The population average kinship coefficients under the four OCS_minKin strategies are 0.22, 0.23, 0.25, and 0.32, respectively.

### 3.3. QTL Effect Variance and Average Observed Heterozygosity

In this study, we found the inbreeding level of populations has been improved to varying degrees due to the existence of selection in breeding strategies. The QTL effect variances and the average observed heterozygosity have also shown corresponding decline, as shown in Table 1.

The values of the QTL effect variances and the average observed heterozygosity of population at the last generation under different breeding strategies have a certain correlation with the population average kinship coefficient. In general, the higher the population average kinship coefficient, the lower the QTL effect variances and the average observed heterozygosity.

## 4. Discussion

Linear programming was previously used as an optimal strategy to select candidates under many constraints in livestock breeding [33]. Several studies have used linear programming to select candidate bulls, which can significantly reduce breeding costs [34] and maximize population breeding goals [35]. Moreover, linear programming has also been utilized to optimize mating allocation, which can directly obtain specific mating combinations. By adding genome information, feeding cost and other parameters, the LP method can help breeders achieve a multi-objective breeding program [36,37,38].

The OCS method can maximize the weighted genetic value of parents while limiting the relationship among them [39], thereby maximizing the genetic gain in offspring populations and minimizing the inbreeding level of offspring while maintaining the genetic diversity of the population. Gourdine et al. [40] reported that the OCS method can achieve genetic progress similar to truncated selection in a small local pig population using the simulation method, while the inbreeding level decreased by ~4%. In addition, the OCS method can also be used to maintain genetic diversity and conservation [24,41].

The genetic gain in populations under the LP strategy, OCS strategies and TS strategies gradually increased with generations. The reason is that the parents under these breeding strategies have been selected according to specific selection criteria, which is similar to previous analyses [42]. 

In the LP strategy, breeding candidates are selected according to individual WSG. The goal of the linear programming problem is to maximize the selection index of the offspring population, thus the genetic gain in the offspring population is higher than that of the parent population, which is the same as the previous research conclusions [17]. The average WSG of the 1021st generation population under the LP strategy is 3.41, which is higher than the TS−I strategy (3.36), indicating that the LP method has certain advantages in improving genetic gain.

Under the LP and TS−I strategies, the weight of each trait in WSG has a higher effect on genetic gain than the heritability of the trait. Under the breeding strategy, the final genetic gain in the traits with high weight is higher than that of the traits with low weight. When the weights of the two traits are the same, the trait with higher heritability obtains higher genetic gain than the trait with lower heritability. A recent study [30] has pointed out that the higher heritability can improve the accuracy of GEBV and ultimately achieve high genetic gain, which is contrary to our results; thus, the weight of trait in WSG may play an important role in genetic gain. 

The genetic gain under all OCS strategies is lower than that under TS−II, which is similar to previous studies [21,31]. Improving genetic gain and reducing inbreeding level are two conflicting goals in breeding work. The OCS method can reduce population relationships, which will inevitably come at the cost of losing certain genetic gain. However, the gap between the genetic gain in the population under the OCS strategy and the TS−II strategy is gradually narrowing over the last few generations. This is because under the TS−II breeding strategy, the linkage disequilibrium between markers changes as the population’s inbreeding level increases, resulting in more loss of genetic diversity under this strategy and a decrease in the growth rate of genetic gain. 

There are also differences in genetic gain among different OCS strategies. Under the OCS_minKin strategies, the increase in breeding value of offspring compared to that of parents is greater than the lower bound of breeding value growth, and we found that the higher the lower bound, the higher the final genetic gain. The genetic gain under OCS_maxBVE is maintained at a moderate level.

During the breeding process, the candidate parents for breeding are individuals with high genetic performance; thus, the average kinship of the population increases from generation to generation.

The average kinship of the population under LP strategy is higher than TS−I strategy. This may be due to the following points: (1) the objective function of linear programming is constructed based on maximizing the selection index of the offspring population, which leads to mating between individuals with high genetic performance, and the high kinship between these selected individuals may lead to the increase in the relationship among the offspring population; and (2) in the breeding process, the punishment for inbreeding is not sufficient.

The average kinship coefficient of the population under OCS strategies was lower than the TS−II strategy, indicating that the OCS method can effectively control the inbreeding level during the breeding process. 

The population under the OCS strategy with the goal of maximizing BVE of offspring population has the lowest population kinship coefficient. The population kinship coefficient under the OCS_minKin strategies are lower than the TS−II strategy and higher than the OCS_maxBVE strategy. Among these breeding strategies, the higher the lower bound of breeding value growth, the higher the population kinship coefficient under the breeding strategy.

The QTL effect variance and average observed heterozygosity are used to reflect changes in the genetic diversity of populations during the breeding process. Genetic diversity plays an important role in species’ ability to adapt environmental changes, breeding strategies for crops and farm animals, and protection of endangered species [43,44]. The loss of genetic diversity can lead to a decrease in the population’s ability to respond to selection pressures [45]. Therefore, detecting changes in genetic diversity within a population is of great significance for the design of a breeding strategy [46].

Also, there is a correlation between the QTL effect variance and average observed heterozygosity and the population average kinship coefficient: the higher the final average kinship coefficient of the population, the lower the QTL effect variance and average observed heterozygosity. This also indicates that genetic diversity of population decreases with the inbreeding level increases. Under OCS strategy, population genetic diversity is higher than TS and LP strategies; thus, the OCS method is an optimal breeding method that can achieve sustainable genetic gain during the long-term selection.

Further analysis about genetic correlation and multi-trait breeding may be considered for optimization and improvement in breeding strategy. In the current study, genetic correlation of the simulated population was not included in our analysis, and analysis of multiple traits using LP and OCS approaches for long-term selection should be useful in the actual situation. The objective function in the LP method may contain more parameters from economic aspect. Moreover, in the LP breeding strategy, further study may be considered to assess the impact of inbreeding on WSG when punishing the inbreeding level of the offspring population. In the OCS strategy, the feasibility of the OCS method for multi-trait breeding schemes can also be explored in the subsequent analyses. Many selection methods, such as the Weighted Genetic Selection method and the Genetic Algorithm method have been proposed, which also require further exploration in future research.

Under the LP strategy, the population can achieve higher genetic gain, which means that it can create higher economic value in a shorter period of time. At the same time, shorter growth time also reduces feeding costs and disease risks. Therefore, LP strategy is a breeding strategy that can improve economic benefits in the short term. However, considering the inbreeding level of the population, the OCS strategy is a breeding strategy aimed at achieving long-term economic benefits.

This experiment can provide theoretical guidance for breeders to choose appropriate breeding strategies for different breeding goals by integrating the genetic gain and average kinship changes of multiple generations under LP and OCS strategies. If breeders want to quickly improve the genetic gain in high heritability traits in the short term, they can choose the LP strategy. If breeders want to improve the genetic gain in low heritability traits, increasing the weight of this trait in LP breeding strategies is a considerable option. For populations that require long-term breeding, the OCS strategy is more suitable. The OCS strategy can also be used for the protection of endangered species.

## 5. Conclusions

In this study, we performed breeding simulation to evaluate the effect of the LP method and OCS method during long-term selection in cattle breeding. The results showed that the LP method can significantly improve the genetic gain in the population compared with the TS method, especially for the traits with higher heritability and the traits highly weighted in the breeding process. However, the control effect of LP on population inbreeding level is not improved. The OCS method may balance the conflicting breeding goals of improving genetic gain and reducing population inbreeding levels during breeding process. Application of the OCS method is feasible strategy in breeding that can achieve sustainable genetic gain during long-term selection process.

## Figures and Tables

**Figure 1 biology-12-01157-f001:**
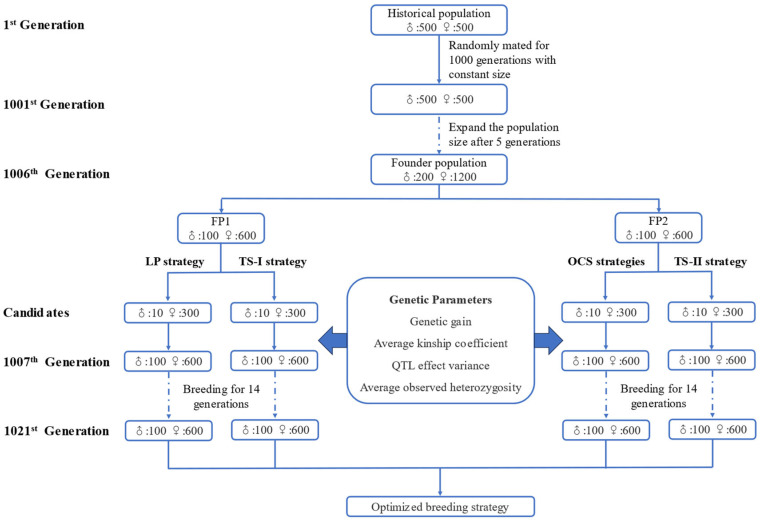
Breeding process of FPs and various breeding strategies.

**Figure 2 biology-12-01157-f002:**
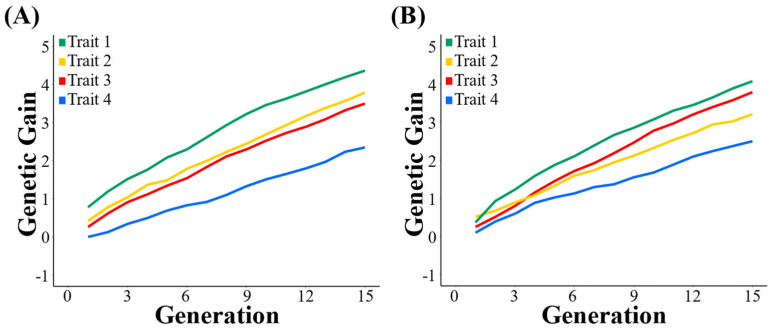
Genetic gains of four traits in populations under TS−I and LP strategies. (**A**) TS−I strategy; (**B**) LP strategy.

**Figure 3 biology-12-01157-f003:**
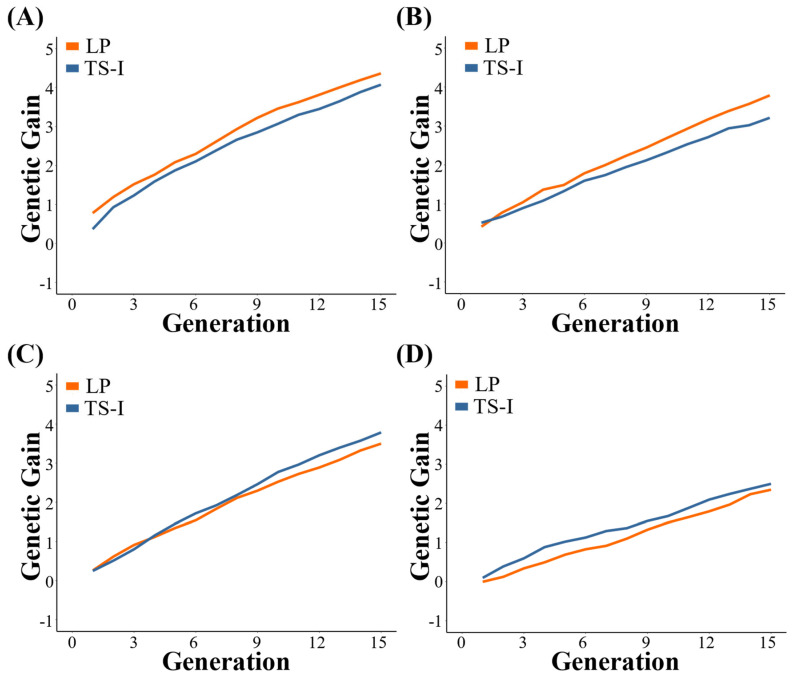
Comparison of genetic gain under TS−I and LP strategies. (**A**) Trait 1; (**B**) Trait 2; (**C**) Trait 3; (**D**) Trait 4.

**Figure 4 biology-12-01157-f004:**
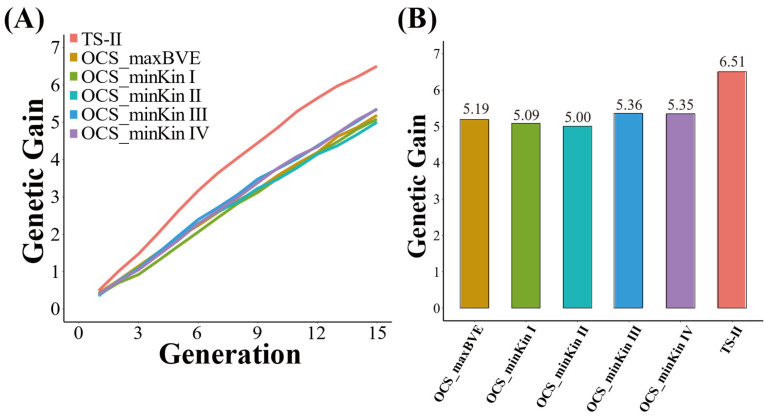
(**A**) Comparison of genetic gain under TS−II and OCS strategies; (**B**) comparison of average kinship under TS−II and OCS strategies.

**Figure 5 biology-12-01157-f005:**
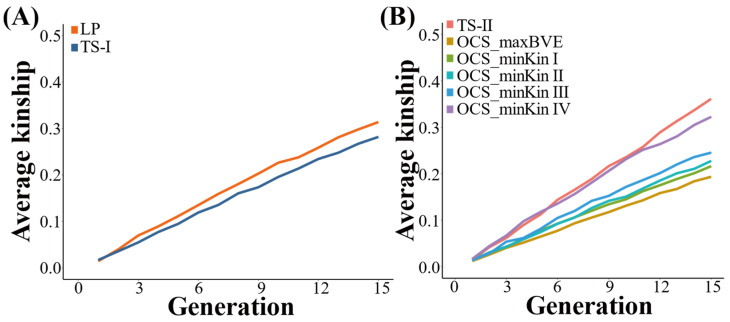
(**A**) Comparison of average kinship under TS−I and LP strategies; (**B**) comparison of average kinship under TS−II and OCS strategies.

**Table 1 biology-12-01157-t001:** The QTL effect variances and the average observed heterozygosity of the FP and the 1021st generations of all breeding strategies.

Breeding Strategy	$\sigma^{2}$		H_O_	
	FP	G1021	FP	G1021
TS−I	0.00486	0.00324	0.21042	0.15191
LP	0.00493	0.00344	0.21042	0.14664
TS−II	0.00486	0.00145	0.21042	0.13712
OCS_maxBVE	0.00493	0.00278	0.21042	0.17097
OCS_minKin I	0.00493	0.00274	0.21042	0.16668
OCS_minKin II	0.00493	0.00260	0.21042	0.16418
OCS_minKin III	0.00493	0.00248	0.21042	0.15949
OCS_minKin IV	0.00493	0.00215	0.21042	0.14429

## Data Availability

The datasets used and analyzed during the current study are available from the corresponding author upon reasonable request (LYX). The data are not publicly available to preserve the privacy of the data.

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
