# Peer review of "Evaluation of Linear Programming and Optimal Contribution Selection Approaches for Long-Term Selection on Beef Cattle Breeding"

_biology, 2023, doi:10.3390/biology12091157_

Round 1
Reviewer 1 Report
The purpose of this study are to explore the application effect of LP method in controlling inbreeding level and improving genetic gain in cattle breeding, and also evaluate the effects of OCS strategies with different breeding objectives during long-term selection. The material used in the research is sufficiently numerous, the description in Materials and Methods chapter are correctly. The results are described correctly. The discussion is exhaustive. Summary of the results are correct. Some corrections are needed. The proposed changes are listed below.
General comments:
Please prepare the article in accordance with the instructions for authors:
The references chapter for page ranges use long () from the symbol function, instead of short (-) from the keyboard
In References chapter please use a "dot" after each abbreviation, for example J. Dairy Sci.; Ital. J. Anim. Sci.
Detailed Comments:
L45
Are there any other effects (for example, deterioration of reproductive parameters, more frequent diseases) with the increase in inbreeding. Are there any general rules for selecting individuals for breeding in order to reduce the inbreeding level
L64 Hjorto et al. [24] according to Reference section
L67 Kohl et al. [25]
L68 delete [25]
L81 What is the significance of this research for breeding work on beef cattle?
L144 2.2.3. dot after "3"
L216 Trait 1, Trait 2, Trait 3, Trait 4, Trait 5 , space after "Trait" throughout main article
L231 Figure 1; Figure 2, etc space after "Figure"
Figure 4B add the values above the bars
L295 Goutdine et al. [40]
L297 delete [40]
Table 1 What was the value of inbreeding coefficients (relationship) in this population for the compared breeding strategies
L353 4.4. dot after "4"
L427 "BMC Genet." instead of current form
L469 Math. Bios. instead of current form
L496 "Fundamental concepts genetics the genetics of inbreeding depression"
Author Response
Responses to reviewer 1
General comments:
The references chapter for page ranges use long (- ) from the symbol function, instead of short (-) from the keyboard
AU: Revised. Please see Reference chapter.
In References chapter please use a "dot" after each abbreviation, for example J. Dairy Sci.; Ital. J. Anim. Sci.
AU: Revised. Please see Reference chapter.
Detailed Comments:
L45
Are there any other effects (for example, deterioration of reproductive parameters, more frequent diseases) with the increase in inbreeding. Are there any general rules for selecting individuals for breeding in order to reduce the inbreeding level
AU: Previous studies have suggested that the fitness traits, such as survival, reproduction, and disease resistance, are more susceptible to inbreeding, leading to a decrease in the phenotypic values of these traits. Reducing the co-ancestry and the genetic relationship between mating individuals has always been recommended as a principle to reduce the level of inbreeding in offspring, as it can decrease variation of each ancestor’s contribution.
L64 Hjorto et al. [24] according to Reference section
AU: Revised.
L67 Kohl et al. [25]
AU: Revised.
L68 delete [25]
AU: Revised.
L81 What is the significance of this research for breeding work on beef cattle?
AU: Revised.
L144 2.2.3. dot after "3"
AU: Revised.
L216 Trait 1, Trait 2, Trait 3, Trait 4, Trait 5 , space after "Trait" throughout main article
AU: Revised.
L231 Figure 1; Figure 2, etc space after "Figure"
AU: Revised.
Figure 4B add the values above the bars
AU: Revised.
L295 Goutdine et al. [40]
AU: Revised.
L297 delete [40]
AU: Revised.
Table 1 What was the value of inbreeding coefficients (relationship) in this population for the compared breeding strategies
AU: In our study, we calculated the average kinship coefficient of each generation population to represent the inbreeding level of that generation.
L353 4.4. dot after "4"
AU: Revised.
L427 "BMC Genet." instead of current form
AU: Revised.
L469 Math. Bios. instead of current form
AU: Revised.
L496 "Fundamental concepts genetics the genetics of inbreeding depression"
AU: Revised.

Reviewer 2 Report
Comments for the authors
Major comments
Material and methods
- You should report the approval number of this study by an ethical committee of your institute or university.
- L84: give more details about the selected population (e.g., breed, age)
Discussion
- You could remove the subtitles from this section. It is unusual to use subtitles in ‘’discussion’’.
- You could add a paragraph for the economic impact of your results in livestock management.
Minor comments
- L48-49:.. the Linear programing method (LP) and the Optimal 48 contribution selection method(OCS)
- L50: .. steps for the application
- L59: .. under the OCS strategy
- L95: .. expanded in the breeding process
- L147: .. genetic contributions to the offspring population..
- L209: .. we utilized the simulation method to construct breeding populations in 209 cattle and conducted a breeding program..
- L217: .. increase with generations, as shown in Figure 2
- L315: .. pointed out that the
- L343: .. was lower ..
- L354: .. in the genetic diversity of..
Author Response
Responses to reviewer 2
Major comments
Material and methods
-You should report the approval number of this study by an ethical committee of your institute or university.
AU: In our study, real animal individuals were not involved as experimental materials. Our study was finished based on a simulated population, so there are no Institutional Review Board Statement or approval number.
-L84: give more details about the selected population (e.g., breed, age)
AU: Thanks for your suggestion. In our study, the population was generated based on the general genotype and phenotype information of beef cattle, and did not refer to a single breed. Age is not a variable in our study, so in the breeding process, we use generations as units and do not overly consider individuals’ age.
Discussion
-You could remove the subtitles from this section. It is unusual to use subtitles in ‘’discussion’’.
AU: Revised. Please see Discussion chapter.
-You could add a paragraph for the economic impact of your results in livestock management.
AU: Revised. Please see Line 382-387.
Minor comments
- L48-49:.. the Linear programing method (LP) and the Optimal 48 contribution selection method(OCS)
AU: Revised.
- L50: .. steps for the application
AU: Revised.
- L59: .. under the OCS strategy
AU: Revised.
- L95: .. expanded in the breeding process
AU: Revised.
- L147: .. genetic contributions to the offspring population..
AU: Revised.
- L209: .. we utilized the simulation method to construct breeding populations in 209 cattle and conducted a breeding program..
AU: Revised.
- L217: .. increase with generations, as shown in Figure 2
AU: Revised.
- L315: .. pointed out that the
AU: Revised.
- L343: .. was lower ..
AU: Revised.
- L354: .. in the genetic diversity of..
AU: Revised.

Reviewer 3 Report
The authors aimed to evaluate the importance and the usefulness of linear programming and optimal contribution of selection approaches for the long-term selection of cattle.
The manuscript can be published but careful revision must be carried out as detailed below.
Major issues
2.1. The authors should extend this sub-section. This is the heart of the study and as it is now, only a summary is presented. The authors must revise to present all the fine details of the work performed.
2.2. A detailed description of the method followed to design these two selection types must be included. Also, a justification of these two selection types must be included. The justification must include the other types of selection that were considered but were rejected and the advantages of the selection types followed at the end.
Minor issues
The objectives of the study must be defined more clearly.
In the conclusion, please do not introduce new ideas, but rather extend on the findings of the study performed.
Author Response
Responses to reviewer 3
Major issues
2.1. The authors should extend this sub-section. This is the heart of the study and as it is now, only a summary is presented. The authors must revise to present all the fine details of the work performed.
AU: Revised. Please see Part 2.1.
2.2. A detailed description of the method followed to design these two selection types must be included. Also, a justification of these two selection types must be included. The justification must include the other types of selection that were considered but were rejected and the advantages of the selection types followed at the end.
AU: We have clarified the detailed description of LP and OCS methods in the introduction, please in line 48-70; the justification of method was evaluated by genetic parameters including genetic gain, the average kinship coefficient, QTL effect variance and average observed heterozygosity across generations. In this study, we did not include more other method including Weighted Genetic Selection method, Genetic Algorithm method, etc.. Yes, these methods may also provide value able insights for future study. We have added this limitation in discussion part.
Minor issues
The objectives of the study must be defined more clearly.
AU: Revised. Please see Introduction chapter.
In the conclusion, please do not introduce new ideas, but rather extend on the findings of the study performed.
AU: Revised. Please see Conclusion chapter.

Round 2
Reviewer 3 Report
The manuscript has been improved and is almost ready for acceptance.
If you can please include a brief passage in the discussion or the conclusions regarding the clinical applications of the findings, that will be ideal.
Thank you.
Author Response
Responses to reviewer 3 (Round 2)
-The manuscript has been improved and is almost ready for acceptance.If you can please include a brief passage in the discussion or the conclusions regarding the clinical applications of the findings, that will be ideal.
AU: Revised. Please see the Line 387-394, 403-404.
